# Immunocapture of cell surface proteins embedded in HIV envelopes uncovers considerable virion genetic diversity associated with different source cell types

Sarah Sabour[1,2], Jin-fen Li[2], Jonathan T. Lipscomb[2], Ariana P. Santos Tino[3], Jeffrey A. Johnson[2]*

1 ORISE Fellowship Program, Oak Ridge, Tennessee, United States of America, 2 Division of HIV Prevention, CDC, Atlanta, Georgia, United States of America, 3 DESA Group, Columbia, South Carolina, United States of America

* jjohnson1@cdc.gov

## Abstract

HIV particles in the blood largely originate from activated lymphocytes and can overshadow variants which may be expressed from other cell types. Investigations of virus persistence must be able to distinguish cells refractory to viral clearance that serve as reservoirs. To investigate additional cell types that may be associated with *in vivo* HIV expression we developed a virus particle immunomagnetic capture method targeting several markers of cellular origin that become embedded within virion envelopes during budding. We evaluated the ability of markers to better distinguish cell lineage source subpopulations by assessing combinations of different antibodies with cell-sorted *in vitro* culture and clinical specimens. Various deductive algorithms were designed to discriminate source cell lineages and subsets. From the particle capture algorithms, we identified distinct variants expressed within individuals that were associated with disparate cellular markers. Among the variants uncovered were minority-level viruses with drug resistance mutations undetected by sequencing and often were associated with markers indicative of myeloid lineage (CD3⁻/CD10⁻/CD16⁺ or /CD14⁺, and CD3⁻/CD16⁻/CD14⁻/CD11c⁺ or /HLA-DR⁺) cell sources. The diverse HIV genetic sequences expressed from different cell types within individuals, further supported by the appearance of distinct drug-resistant variants, highlights the complexity of HIV reservoirs *in vivo* which must be considered for HIV cure strategies. This approach could also be helpful in examining *in vivo* host cell origins and genetic diversity in infections involving other families of budding viruses.

## Introduction

The study of human immunodeficiency virus type 1 (HIV-1, HIV) persistence *in vivo* is complex, engendering multifaceted examinations of susceptible cellular lineages that exist in blood and anatomical sites. Achieving durable suppression to undetectable virus levels and aims for

**Data Availability Statement:** Sequences are submitted to GenBank (accession nos. OR359764-OR359776).

**Funding:** The author(s) received no specific funding for this work.

**Competing interests:** The authors have declared that no competing interests exist.

a functional cure require successful inhibition of all cell reservoirs of HIV that support infectious virus production [1–3]. The clinical examination of HIV, however, is typically performed on peripheral blood plasma and cells which do not encompass organ-resident cells. Therefore, blood sampling does not adequately represent the sources contributing to the *in vivo* viral quasispecies. Moreover, the inability to distinguish virus by source cell type creates an obstacle to identifying reservoirs that are key to persistence.

The difficulty in sampling HIV in tissues such as the lungs, genital tract, central nervous system, and immune organs like the spleen, presents a considerable barrier to suitably studying foci of viral persistence. Within these compartments, target cells permissive to HIV infection might include resident T cells (lymphocytes), macrophages (Mphage), monocytes (Mc), dendritic cells (DC) and natural killer (NK) cells, to name only a few, each having varying levels of infection permissiveness that are dependent on developmental and activation stages [4–14]. Additionally, preclinical studies and tissue pharmacodynamic analyses have shown that antiretroviral (ARV) drug penetrance into anatomical sites varies with drug class, as does drug permeability and metabolism by the resident permissive cells [15, 16]. These factors can alter the amount of active drug [17, 18] in an infected cell and, thereby, the extent of viral suppression.

Since HIV establishes life-long infection involving a myriad of *in vivo* reservoirs, identifying tissue sources of occult virion expression is important to understanding HIV persistence in the presence of highly effective treatment. An earlier investigation by others explored identifying sources of HIV by targeting cellular proteins that incorporate into virion envelopes. They found that by using monoclonal antibodies (mAb) against incorporated cell membrane cluster of differentiation (CD) proteins, particles could be largely separated by macrophage-origin (CD14, CD36, and CD68) and lymphocyte-origin (CD3 and CD26) markers [19, 20]; however, these markers are not necessarily lineage specific. Moreover, this work examined persons co-infected with *Mycobacterium tuberculosis* (Mtb), which enhances both the contribution of macrophage-derived HIV and the ability to capture virion from those sources. In the majority of infections HIV expressed from macrophage comprise less than 1% of the viral swarm [16]; therefore, the previous investigation could have resulted in an atypical, enhanced detection of particles from myeloid lineage cells. Another significant limitation to this earlier study was that only two broadly defined cell types were investigated with no discrimination deeper into cell lineage, thus, it was quite limited in the ability to examine cell source types [21–27].

Because of the difficulty in accessing tissue-resident cell sources to assess the effectiveness of suppressive treatment modalities within compartments, we expanded on the concept of targeting incorporated host cell proteins for studying virions of the peripheral blood swarm. Immunomagnetic capture algorithms were designed that target a variety of discriminating host cell membrane CD proteins to isolate HIV particles originating from a wider representation of immune system cell types. In deductive sequences, biotinylated mAb bound to streptavidin magnetic beads isolate virion particles associated with specific host-cell associated CD proteins that incorporate in virus envelopes as a result of the budding process.

We designed and applied this assay to gain greater awareness of the cell types that yielded HIV populations in blood and examined the genetic sequences of those variants. We found many viruses, including those with drug resistance mutations, were at low relative frequencies that were obscured from detection using typical sequencing methods. Hence, an important feature of targeting expressed virions rather than cells is that their source cell types could be elucidated even if they are sequestered within tissues with only trace virus representation in peripheral blood. The capacity to identify miniscule, yet potentially important, reservoirs which are refractory to therapy is key for approaches aiming to improve viral suppression and for strategies to achieve functional cure or eradication.

## Methods

### Protection of human subjects and specimen source populations

Laboratory experiments on remnant clinical plasma specimens were carried out in accordance with Health and Human Services and Office of Human Subjects Protection guidelines on the use of humans in research. The specimens were previously obtained [31–33] under approval of the University of the Witwatersrand's Human Research Ethics Committee (Medical) and Centers for Disease Control and Prevention Institutional Review Board (IRB) protocols (CDC IRB protocol nos. 1774, 3621, and 3910) and received CDC project determination approval (approval received 9/13/2013, specimens accessed 9/16/2013) to be used in the present study. The approved protocols included written consent for drug resistance testing, storage, and future HIV testing. Specimens were previously coded and individual identifying information is indefinitely withheld from CDC and authors. In designing and verifying the assay performance numerous remnant clinical samples (n>400) from the previous studies were used in addition to 10 commercial seroconversion panels (Zeptometrix), and three fresh leukopak donations.

In the present report demonstrating the particle capture assay for examining the diversity of HIV variants in blood, the plasma of nine individuals with different clinical histories were used. To increase our ability to identify HIV genetic variants between cell source types, we selected from characterized specimens that we previously found had antiretroviral drug resistance mutations detected by bulk or sensitive genotyping. Drug resistance mutations arise from selective antiretroviral drug pressure and so serve as markers of genetic evolution beyond the polymorphisms that naturally occur due to the mutation rate of the viral reverse transcriptase.

### Immunomagnetic virion capture assay

We redesigned and expanded an earlier concept [19] of virus particle immunocapture to create algorithms for deeper investigation of HIV variants expressed *in vivo*. Immunocaptures used streptavidin-coated magnetic microbeads (μMACS, Miltenyi Biotech, San Diego, CA) and in-house biotinylated (BTAG, Sigma) monoclonal antibodies (mAbs) to different cell markers associated with cell types reported in the literature to be permissive of HIV infection. In all, 17 mAbs (Santa Cruz Biotechnology, Inc) were evaluated in an iterative process further described in the Results section. Cell sources of HIV were assumed to have been CD4-positive as a requirement for efficient infection [28]. All mAb were concentrated from their stock 0.2 mg/mL to 2 mg in 100 μL PBS prior to biotinylation using Amicon Ultra-0.5 centrifugal filter devices. Four μL (2 μg) of the first biotinylated Ab in the series was added to 100 μL of streptavidin-coated microbeads then incubated for 10 minutes at room temperature (21–25 ˚C) on a roller platform. This amount of biotinylated Ab was expected to saturate the streptavidin beads assuming each 150–160 kDa Ab was labeled with 10 biotin molecules on average (100 μL of beads will bind up to 1.6 μg biotinylated mAb). The amount of antibody-bead material used in each column was sufficient to support binding of up to 500,000 virions, in that adding more virus decreased specificity of binding. The bead-Ab complex was centrifuged at 8,000 rpm (6,000 x*g*) for 10 mins and washed twice in 100 μL PBS+1% BSA+1% Tween 20 (Millipore-Sigma) to remove non-adsorbed Ab, then blocked overnight (16–20 hours) at 4˚C again in 100 μL PBS+1% BSA+1% Tween 20.

The bead-Ab complex for each capture Ab was prepared as above then resuspended in either 200 μL of blood plasma from persons with HIV or clarified culture supernatants and incubated for 30 minutes at room temperature on a horizontal mixer-roller. After incubation,

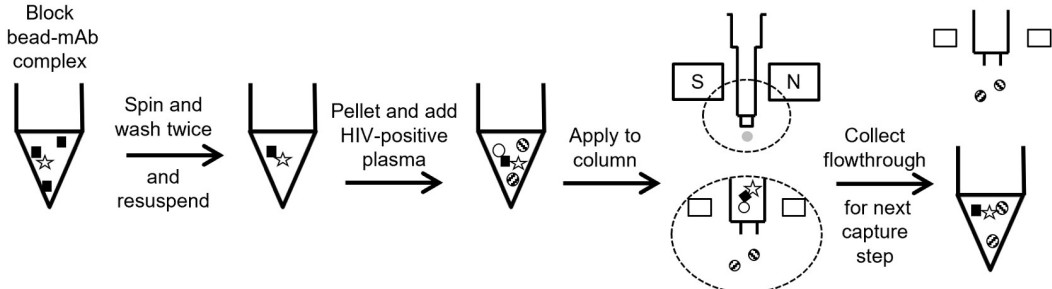

**Fig 1. Overview of the immunocapture assay procedure.** Biotinylated monoclonal antibody (mAb, black box) and streptavidin beads (star) are incubated for 10 minutes at room temperature. Antibody-bead conjugates are pelleted and resuspended in wash buffer; this is repeated twice. The mAb-bead complex was blocked overnight at 4°C in BSA-tween 20 after which HIV-positive plasma is used to resuspend the antibody bead pellet and allowed to incubate for 30 minutes at room temperature. The capture mix is applied sequentially to the μMACS columns. Non-target virions (striped circles) are washed through the column while target virions (open circle) are retained in the column. Lysis buffer is added to the column to lyse target virions and release their RNA. The flow-through retained after each column is applied to the next mAb incubation and capture column in the series and the process repeated.

each bead-Ab-virus complex was applied to a new buffer-equilibrated μMACs column at each step in the algorithm series which is then attached to a magnetic multistand (Miltenyi Biotech) (Fig 1) collecting the flowthrough then adding and collecting 30 μL PBS+2% FBS+1% Tween 20 to displace the column void volume. The columns are then washed three times with 400 μL PBS+2% FBS+1% Tween 20 to remove residual virus that was not bound in the Ab-complex magnetically retained on the column. Different wash buffer formulations were evaluated with surfactant to increase capture specificity as any residual non-specific column retention would lead to undesirable PCR amplification of non-target virus (S1 Table). The wash was also assessed for its ability to preserve intact particle binding on the columns since Tween 20 is reported to be cytolytic (S1 Fig). At each step the wash flow-through was collected to recover unbound viruses then incubated with the next mAb in the series for application to subsequent columns. This process was repeated until all mAbs in the algorithm were incubated and applied. Fig 2 depicts the final capture scheme devised to examine particles in blood plasma. The first column in every series was comprised of beads that were not conjugated with Ab as a negative control and to exclude from further analysis samples in which particles non-specifically adhered to the column matrix.

After washing, the column-bound virions at each step were directly lysed on the column to recover RNA for genetic analysis. For this, 50 μL of AVL lysis buffer from the QIAamp Viral RNA Mini Kit (QIAGEN) was added to the columns, allowing contents to flow out, and then cleared with another 150 μL AVL lysis buffer. An additional 360 μL of lysis buffer was added to the eluate tube of lysed product then incubated at room temperature to complete the 10-minute lysis prior to proceeding with the RNA purification protocol as described in the kit manual. The protocol described in this peer-reviewed article is published on protocols.io and is included for printing purposes as S1 File (also available at dx.doi.org/10.17504/protocols.io. e6nvwdom7lmk/v1).

## Capture assessments with virus expressed from sorted-cell culture

The specificity and relative recovery of virion captures for lymphoid and myeloid-derived particles was examined on cultures of enriched cell subsets sorted from 60 million fresh leukopak cells purchased from a blood donor service. Lymphocyte sorts were first enriched for CD4 cells by magnetic removal of CD8, CD16, CD11b, CD36, CD56, CD123 and CD235a cells

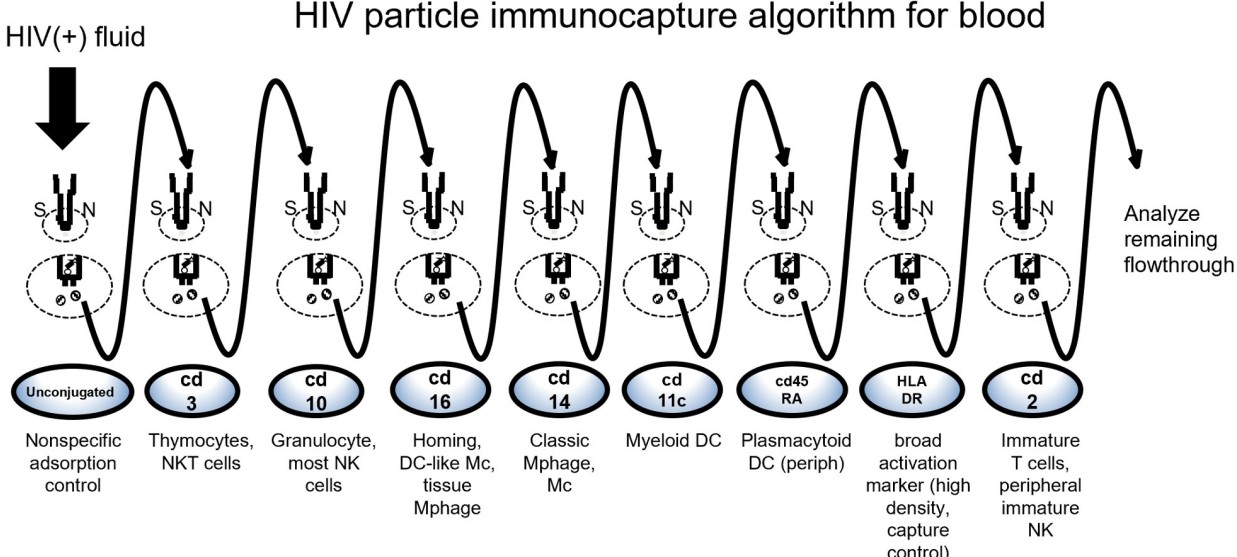

**Fig 2. Primary algorithm for immunocapture of HIV from blood plasma with an emphasis on investigating myeloid origin sources.** The sequential order of antibody captures is shown, and the possible source cell types associated with the targeted marker is indicated below each step. NK, natural killer cell; NKT, natural killer-like T cells; DC, dendritic cell; Mc, monocyte; Mphage, macrophage; periph, peripheral/non-tissue-resident.

(Beckton Dickinson iMag Human CD4 T Lymphocyte Enrichment) then applied to a cell sorter (Sony SH800Z) gated for CD3+ and CD45+. This dual-positive fraction was then generously gated for CD4+/CD8- cells, followed by harvesting CD45RO+ and CD45RA+ fractions. The enriched CD45RO+ and CD45RA+ fractions were separately cultured overnight to rest without cytokine stimulation. The next morning each culture was inoculated with HIV-1$_{IIIB}$ [29] for 3 hours again without cytokine stimulation to avoid altering surface marker expression, then washed of inoculum and maintained in culture until cytopathic effect (CPE) was visualized.

For myeloid cell sorting to examine specificity with virion that incorporated CD16 and CD14 we found the mechanical sorter was detrimental to the viability of these cell subsets. Thus, 10 million leukopak PBMCs were manually enriched using the Human Monocyte Enrichment Set (Becton Dickinson). The depletion cocktail removed CD3, CD45RA, CD19, CD56, and CD235a cells. The monocyte-enriched cells were immediately placed into culture and inoculated with HIV-1$_{ADA-M}$ [30] for two hours, then washed and replenished with media containing tumor necrosis factor (TNF)-α. Cell culture supernatant was collected after 2.5 days when viral RNA was detected. For all cultures, ≤200 μL supernatant (500,000 HIV copies maximum) was used for analysis in the antibody viral capture method. Samples with high viral loads were diluted with PBS +1% BSA.

## RT-PCR amplification of HIV variants

RNA extracts of captured virion were reverse transcriptase (RT)-PCR amplified in *pol* RT regions using SuperScript™ III One-Step RT-PCR System (Invitrogen) using different combinations of primers to overcome possible polymorphisms in primer binding sites as previously described [32]. HIV RNA amplicons were detected by sensitive real-time PCR using Qiagen EpiTect HRM PCR kits containing EvaGreen intercalating dye and Promega GoTaq® qPCR Master Mix containing BRYT Green® Dye. Nested PCR products of the RT regions were

generated using Platinum™ SuperFi™ II PCR System for sequencing. Cultured virus supernatants were treated with DNase for one hour at 37˚C (DNA-free kit, Invitrogen) to remove proviral DNA in solution that was found to cause spurious amplification from column fractions.

## Genetic sequencing

PCR products from capture steps were bulk sequenced in the reverse transcriptase (RT) region (between codons 60–219) using the BigDye Sequencing mix v1.1. Sequencing reaction conditions were as follows, 25 cycles of 96˚C for 10 seconds, 50˚C for 5 seconds and 60˚C for 4 minutes and a hold at 4˚C. Sequences were analyzed using 3130xl or SeqStudio Genetic Analyzers (Applied Biosystems). For a subset of specimens with drug resistance mutations, we also analyzed available total viral RNA extracts from clinical samples using next generation sequencing (NGS) (Illumina MiSeq) for comparison to the genotypes of captured virion fractions. NGS amplicon libraries were created using the Nextera XT DNA Library Prep Kit and analyzed using MiSeq Reagent Kit v2 300-cycle kit. Variant analysis was conducted against an HIV HXB2 RT reference using CLC Genomic Workbench 20.0.3 and a custom variant calling workflow. Sample library input reads were first trimmed using quality scores and adapter trimming. Trimmed reads below 25 bp were discarded. The mapped reads were analyzed for variants using the Low Frequency Variant Detection 2.1 tool.

## Results

### Capture algorithm evaluations

In assessing a best-performance protocol, we found that neither higher amounts of antibody, using more than 100 µL of bead complex, nor extending the time of capture incubation beyond 30 minutes increased virion recovery. Applying greater than 500,000 virion copies/column was found to spuriously yield amplifiable RNA from the Ab-negative primary column due to non-specific column retention. After establishing a column-binding and wash method that alleviated non-specific virion retention, we initially evaluated Abs to 12 CD markers of different cell types associated with macrophages, monocytes, NK cells, dendritic cells and different developmental stages of T cells among others. This was later expanded to 17 CD markers (see S2 Table).

Initial capture approaches including anti-CD15, CD36, CD44, CD55, and CD68, reported by earlier investigators, often yielded genotypes that were highly ambiguous when used within the capture scheme, reflecting insufficient separation of variant subpopulations. This ambiguity was not unexpected given the presence of these markers on several cell types: CD36 (Mphage, platelets, DCs) and 68 (mononuclear phagocytes, granulocytes, low expression on T cell subsets, NK) and CD55 and CD44 are on many lymphoid cell types, the latter being highly expressed in gut tissue. An example of virion heterogeneity associated with CD markers and the consequence of capture order on the genotypes of recovered virus is provided in Fig 3 (GenBank accession nos. OR359764-OR359776). For this example, CD2 and CD44 are lymphoid origin cells, CD36 is myeloid, CD15 can be myeloid or activated T cells. Here we observed that: anti-CD15 cannot distinguish virus from activated T cells versus myeloid origin; CD44$^+$ virion are highly polymorphic and shares subpopulations with CD2$^+$ virus and CD15$^+$ virus; CD36$^+$ virion are distinct from lymphocyte but is too broad (this marker is associated with CD16+ and CD16- macrophage and CD14+ and CD14- monocytes); after removal of CD15$^+$ virion the remnant CD44$^+$ virus population closely matches the CD2 sequence; and the CD2-associated virion genotype is largely preserved in both capture series suggesting a unique subpopulation (CD2 is a marker of immature T cells, NK subsets and lymphoid tissue resident plasmacytoid DCs). In this example there appeared two divergent non-nucleoside RT inhibitor

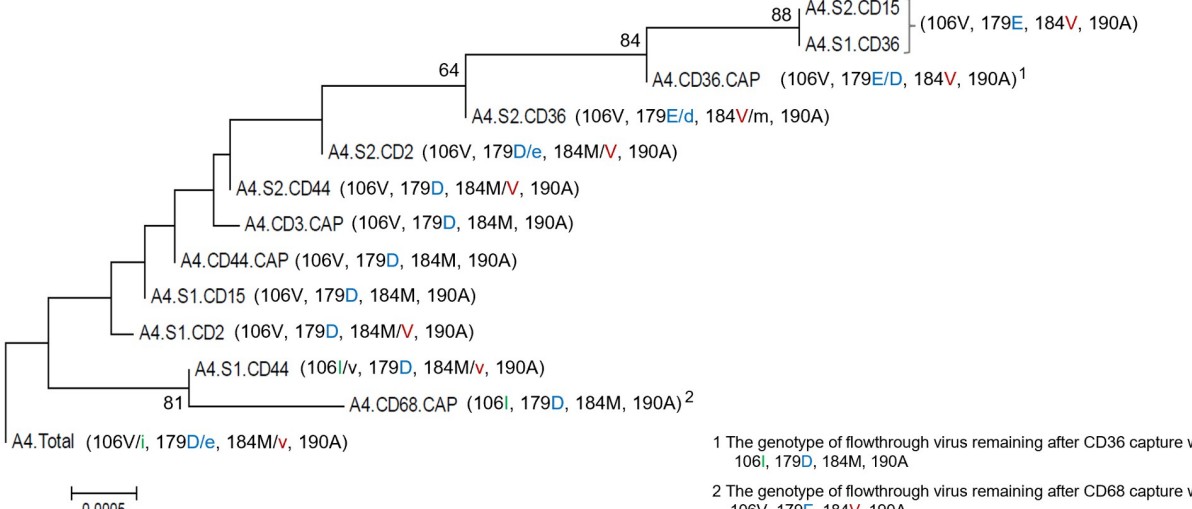

**Fig 3. Example of assessing anti-CD antibodies and effect of altering capture order on the viral sequences obtained from a plasma sample.** The "Total" sequence at the bottom is the bulk genotype of the sample. Series 1 (S1) antibody order was anti-CD2 → CD36 → CD44 → CD15; series 2 (S2) order was anti-CD15 → CD36 → CD44 → CD2. IDs with "CAP" in the title are the sequences obtained when the single capture was applied to the sample to compare to the sequence obtained when the marker was used within a series. CD3 and CD68 are pan lymphocyte and macrophage markers, respectively. Only mutations associated with drug resistance are shown and differences in mutations are highlighted by amino acids in color. The relative proportion of amino acid at codons with mixtures are indicated as majority (capital letter), minority (lower-case letter), or equivalent proportions (both capital letters). Inferred using the Neighbor-Joining method and distances computed using the Tamura 3-parameter method with 500 bootstrap replicates. Rate variation was modeled with gamma distribution (5-parameter) (MEGA v6). Bootstrap values >60 are indicated at nodes.

resistant populations, one acquiring the 179E mutation and those variants acquiring 106I. The direct capture using only the pan-Mphage marker CD68 pulled out an unambiguous V106I variant sequence and could serve as a Mphage catch-all. However, for greater myeloid resolution, as with CD36 above, CD14 and CD16 markers would later be used.

To improve identification of potential virus sources, we explored substituting some of the less discriminatory lineage Abs to include others that may work better in a deductive sequence of cell sources. An algorithm of the order, CD16, CD14, CD31, CD45RA, CD45RO, HLADR, CD27, CD3, CD2, and CD21 (scheme B), was able to better distinguish expressed viral genomes in providing less ambiguous genotypes at each step. Part of the design logic with using HLA-DR mid-sequence, for example, is that it is a marker highly expressed on several key cell types and whose absence can distinguish sublineages, but also it can serve as a capture control marker due to its constitutive abundance. Capturing markers such as CD16, CD14, CD31, CD45RA and CD45RO ahead of the HLA-DR capture narrows the HLA-DR$^+$ capture to cell sources such as myeloid DC or broadly activated cells that were not captured by any of the upstream markers in the series. We also compared CD14 or CD16 as the first mAb to evaluate separation of the CD14$^+$ +/- CD16$^+$ virus subpopulations and found that using CD16 prior to CD14 (targeting CD16+/14- and CD16+/14+ sources versus CD16-/CD14+ sources first) was better able to distinguish genomes of myeloid-derived virion subpopulations.

We found broader investigation of myeloid and lymphoid sources from blood was permitted with slight changes involving ten mAbs of the order, CD16, CD14, CD45RO, CD45RA, CD31, HLA-DR, CD27, CD3, CD2 and CD21 (scheme c). Given the potential complexity of the virion pool and the overwhelming representation of lymphocyte-derived virus which might interfere with the capture of particles derived from non-lymphocyte sources, we also

evaluated alternative capture algorithms. To initially remove lymphoid-derived virions we placed anti-CD3 and CD10 prior to CD16, CD14 and HLA-DR. As a gauge of CD3 removal and myeloid target enrichment, we evaluated the anti-CD3 capture antibody clone on leuko-pak cells to assess the proportion of CD markers present before and after CD3+ reduction (Fig 4).

Comparing algorithms in which the position of CD3 capture was moved to the front of the series altered the variants obtained in subsequent captures that targeted lymphocyte subset markers and allowed further insight into the relative contributions of sources expressing HLA-DR (S2 Fig). In this examination, note the similarities in $CD16^+$ and $CD21^+$ (monocyte, DCs) virion sequences which demonstrate nucleotide differences to those of $CD2^+$ and $CD3^+$ (T cells) virion. For much of our investigation, we had a particular interest in examining myeloid lineage sources of virus found in blood. Hence, we routinely used a modified eight-Ab algorithm, in the order anti-CD3, CD10, CD16, CD14, CD11c, CD45RA, HLA-DR and CD2 (Fig 2), for blood as it provided the greatest ability to successfully capture and distinguish downstream non-lymphoid-derived variant genotypes. Of note, we retained anti-CD2 as it occasionally uncovered sequences that were genetically distinct from $CD3^+$ variants and presumably derived from immature (*e.g.*, intrathymic) lymphocytes or peripheral NK cells.

We found as little as 250 HIV input copies yielded amplifiable virus from HLA-DR and CD3 captures from high-integrity clinical specimens, whereas higher virus copies (often >2000 copies) were required to begin to discern virions identifiable by other markers. The lower limit of particle capture was dependent on the integrity of the specimen, and detection of individual markers varied widely among the clinical cases. Capture diversity from clinical specimens was best achieved or sometimes only possible when using fresh, never frozen, plasma for the analyses.

## HIV genomic RNA analysis from sorted cell cultures

For assessments involving *in vitro* cell-cultured HIV-1, it was important to DNase treat culture supernatants as the large amount of residual proviral DNA present caused spurious virus detection inconsistent with the specific cell origin targeted. Cultures from enriched and sorted CD45RO+ lymphocytes derived from leukopaks yielded virion that were positive for HLA-DR (activated T cells) and CD45RO (T memory) (Fig 5A). The CD45RA-enriched lymphocyte cell culture yielded virions positive for CD45RA and 45RO, demonstrating expression from transition phenotypes, as well as HLA-DR and CD14 (Fig 5B) (see CD45RA/CD14 note in the following paragraph). The CD4 T cell enrichment column removed CD16 but not CD14 cells, therefore some CD14+ cells may have been collected during the RA sorting. No amplification was observed from the antibody-negative columns.

In directly evaluating the myeloid-derived HIV-1$_{ADA-M}$ virus stock we detected $CD14^+$ and $CD16^+$ virion and, surprisingly, virion that were captured using anti-CD3 (Fig 6A). The detection of HIV RNA associated with CD3 suggested either the presence of low-level residual lymphocyte-derived virion in the source cell culture or virion derived from CD3/CD14 complexes (see Fig 4). To remove this apparent lymphoid virion residue for future myeloid capture analyses, we inoculated the HIV-1$_{ADA-M}$ onto myeloid cultures enriched from leukopaks. The subsequent particles expressed from this culture yielded virions with the CD16 and HLA-DR markers, whereas $CD3^+$ virion were no longer detected (Fig 6B). The myeloid enrichment kit also removed CD45RA+ cells suggesting they could have been the source of $CD14^+$ virions in the original virus stock (activated classical macrophage/monocytes are CD14+/CD45RA+). For later myeloid-focused analyses of clinical specimens, the CD45RO step was replaced with CD11c which served as a better capture of DC-derived virion.

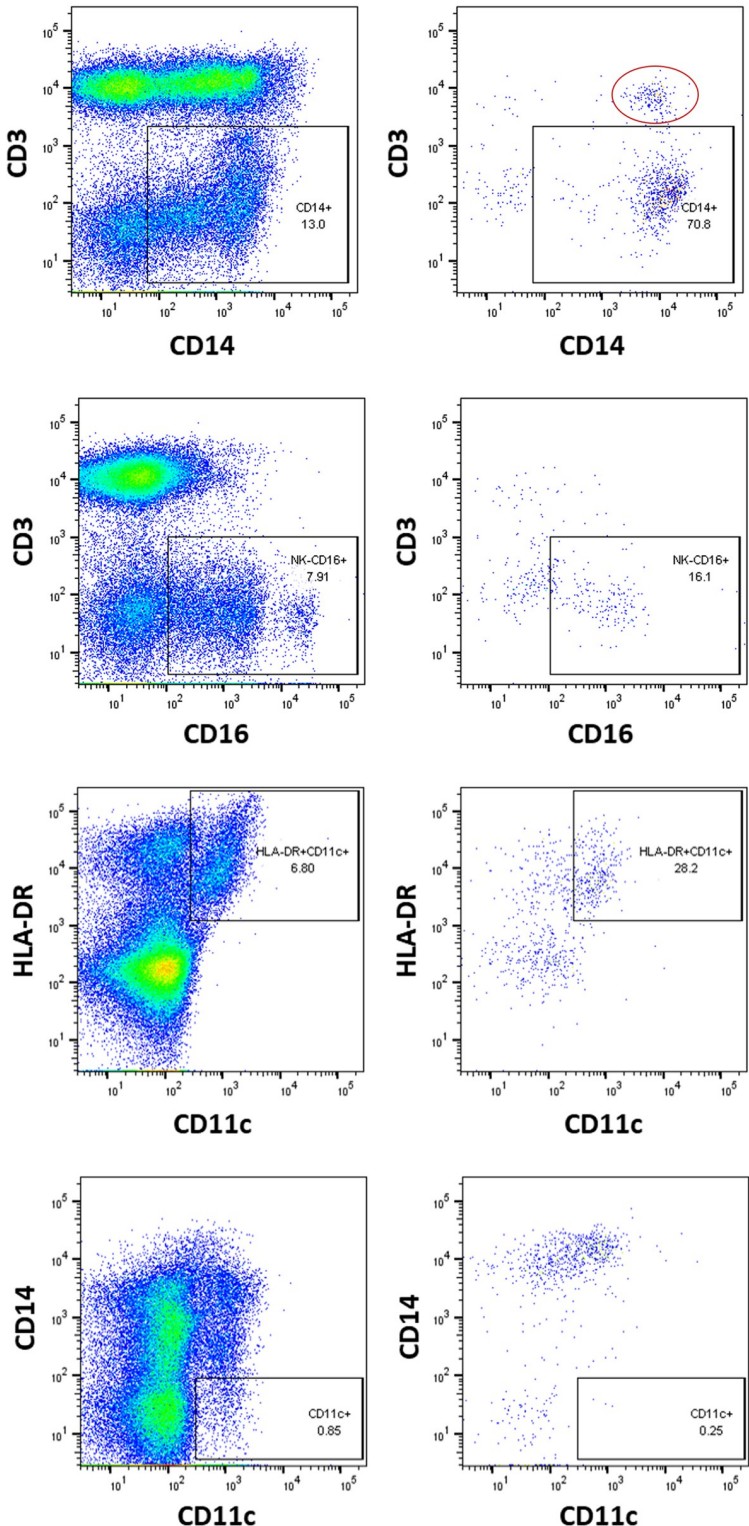

**Fig 4. Use of cell populations to verify subsequent antibody target identification after lymphocyte depletion with the anti-CD3 capture antibody.** Leukopak cells were stained using antibodies to the cell markers shown then analyzed by flow cytometry prior to CD3-depletion ($5x10^6$ cells, left panels) and the proportional increase in CD14, CD16 and CD11c following depletion with anti-CD3 ($5x10^5$ cell input, right panels). The bottom right CD14:CD11c panel distinguished the classical Mc/Macrophage from myeloid DCs. The red oval in the upper-right panel demarcates CD3 +CD14+ circulating large T cell-monocyte complexes that remained after the CD3(-) enrichment [31].

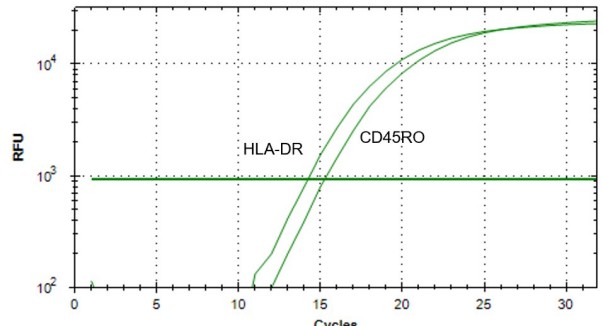

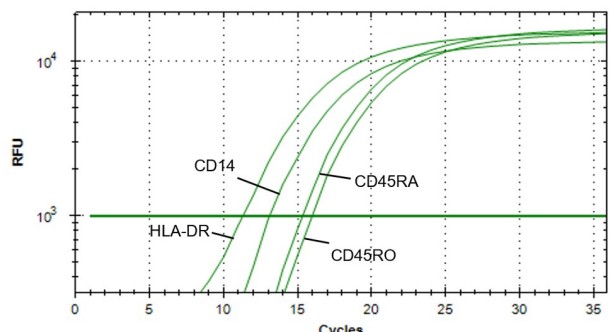

**Fig 5.** Demonstration of amplifiable virion RNA obtained from HIV-1$_{IIIB}$ inoculated onto cultures of CD3$^+$/CD4$^+$ fresh leukopak cells that were sorted for CD45RO+ (A.) or CD45RA+ (B.) co-expression. A. The HIV-1$_{IIIB}$ culture on CD45RO+ cells yielded detectable CD45RO$^+$ and HLA-DR$^+$ HIV RNA. B. The CD45RA-enriched cell culture yielded virions positive for both CD45RA and 45RO as well as CD14 and HLA-DR. The CD4 T cell enrichment column used removes CD16 but not CD14 cells. CD45RA$^+$/CD14$^+$ activated classical monocytes may be acquired in the CD45RA+ sort. No amplification was observed from the antibody-negative columns.

## Segregation of diverse intra-participant HIV variants from blood plasma

As shown in Table 1, applying the algorithms to the blood plasma of individuals with disparate clinical experiences resulted in recovering distinct virion genotypes associated with different cell markers. The immunocapture analyses permitted some key observations on the dynamic expression of variants in response to clinical interventions which are described in the following paragraphs. Differences in drug resistance mutations were segregated among the cell source types that were divergent from the majority virus genomes. Moreover, background polymorphisms not associated with drug resistance were seen to differ at multiple nucleotide positions between source cell types.

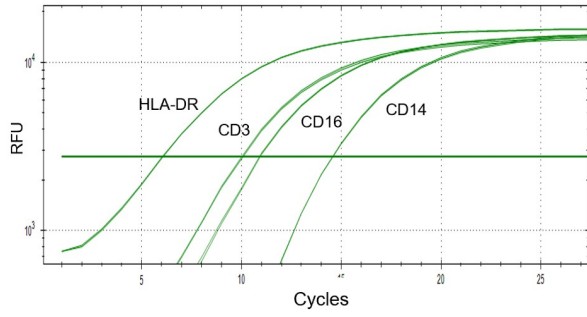

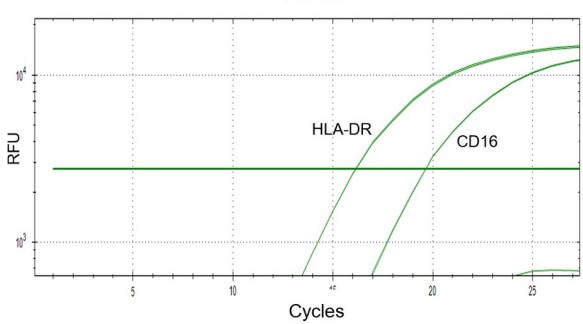

**Fig 6. Demonstration of myeloid-derived virion capture specificity from DNase-treated culture supernatants.** A. Virus captures of the HIV-1$_{ADA-M}$ virus stock revealed that in addition to virion originating from CD16+ and CD14+ cell sources it also contained HIV incorporating the CD3 marker (lymphoid-derived). B. When fresh leukopaks were enriched for myeloid cells then inoculated with the HIV-1$_{ADA-M}$ stock the cultures yielded virions positive for CD16 (migratory/non-resident monocytes) and HLA-DR (myeloid DC) but no CD3$^+$ virions. The myeloid enrichment kit removes CD45RA+ cells suggesting they could have been the source of CD14+ virions in the virus stock (activated classical monocytes are CD14 +/CD45RA+). No amplification was observed from the antibody-negative columns.

**Table 1. Differences in intrapatient nonsynonymous mutations among virions captured from the blood plasma of individuals with various clinical experiences.**

| | | |
|---|---|---|
| Post intrapartum single-dose nevirapine | | |
| 11B | CD16 | K103K*N*, G190*AS* (D123G, V179VI) |
| | CD14 | G190*AS* (D123G) |
| | CD31*[a] | Y188*C* (A129AV) |
| | CD45RA | K103*N* (D123G, Q207G) |
| | HLA-DR | K103*N* (D123G) |
| 26B | CD55*[b] | K103*N* (Q174K) |
| | HLA-DR | Y181*C* (Q174K) |
| | CD45RA | K103*N* (K166R) |
| HIV RNA-positive, pre-seroconversion (SC) series ID 9012 | | |
| 12d pre-SC | CD16 | WT (D123E, D177G, K201R, R211K) |
| | HLA-DR | D67D*N*, T69*N*, K70K*R*, F77F*L*, K101K*HNQ*, G190G*A*, K219K*Q*[1] |
| | CD21*[c] | V90I, V179D[3] |
| 5d pre-SC | CD16 | D67*N*, F77*L*, K101*H*, G190*A*, K219*Q*[2] |
| | CD14 | WT (K104E, D123E, T200I) |
| | CD45RA | WT (K66KR, D121DG, S162CG, L187FL, I202IV, R211K) |
| | CD45RO | V90I, V179D |
| | HLA-DR | WT (D123E, Y144C, D177G, K201R, R211K) |
| | CD21 | D67*N*, F77*L*, K101*H*, G190*A*, K219*Q*[2] |
| Antiretroviral therapy-experienced and unsuppressed | | |
| 6 | CD16 | WT |
| | CD31 | K103K*N* (K104KR) |
| | CD45RA | WT (V60VI, G196GR) |
| | CD3 | K103K*N* |
| | HLA-DR | WT |
| | CD27*[d] | K103*N* |
| 7 | CD3 | WT (Q174E, D185DG) |
| | CD10 | V106*I* (D123E, T131I, K173E, D177E) |
| | CD11c | WT |
| 9 | CD16 | K103K*N*, G190G*A* (K166KR, E169D) |
| | CD14 | K103K*N*, Y181Y*C* (K166KR, E169D) |
| | CD45RO | WT |
| | HLA-DR | K103K*N* (K166KR, E169D) |
| | CD3 | K103K*N* (K166KR, E169D) |
| Acute drug resistance transmission, post-seroconversion | | |
| SH 874 | CD16 | M184*V* (S68SN) |
| | CD10 | M184*V* |
| | HLA-DR | M184*IV* |
| | CD27 | M184*V* |
| | CD31 | M184M*IV* (P176PS) |

*From an earlier capture iteration that would include

[a]recently emergent naïve T cells,

[b]monocytes,

[c]resident dendritic cells,

[d]Tmemory/resident NK. WT, wildtype HIV sequence; **bold italics**, antiretroviral drug resistance mutations;

(mutations in parentheses) are backbone polymorphisms that differed in at least one of the captures;

[1]had high sequence ambiguity;

[2]secondary seeding of an unambiguous variant;

[3]polymorphic variant that appears in different cell populations at 5 days pre-seroconversion.

Analysis of plasma virion from two women provided intrapartum single-dose nevirapine (IDs SDN 11B and 26B), whom we earlier reported had developed a high frequency of drug resistance following this intervention [31], uncovered marked evidence of microenvironmental evolution. The rapid emergence of drug resistance by 6–10 weeks after dosing was now revealed to be a result of simultaneous selection of different drug-resistant variants emanating from discrete cell types (Table 1). Noticeably, selection of divergent resistance genotypes was detected even within myeloid lineage sources during this early post-intervention period.

From the ID 9012 early-acute infection, virus captures of the sample collected when HIV-1 RNA was first detected 12 days pre-seroconversion yielded highly polymorphic genomes, including multiple drug resistance mutations that we had earlier reported from a clonal analysis of the specimen [33]. From the captures performed at the time (scheme B) we found very few yet distinct source cell markers contributing to the virus pool, with HLA-DR$^+$ virion (from highly activated cells) predominating at the earliest timepoint. The captured virion genotypes from this specimen were compared to an earlier NGS analysis of this timepoint and identified two more resistance variants not uncovered by NGS. (S3 Fig). Of note, in scheme B the HLA-DR$^+$ pool would have been heavily represented by lymphocyte-derived virion. From the sample collected one week later, at five days pre-seroconversion, the HLA-DR$^+$ virus was now comprised of non-ambiguous wildtype sequence that had largely replaced the transmitted diverse multi-drug resistant HLA-DR$^+$ virion population. Conversely, the initial drug resistance sequences observed in the earlier transmitted HLA-DR$^+$ virions were now seen associated with other cell source types, namely migratory (CD16$^+$) monocytes and CD21$^+$ cells, the latter a marker associated with tissue-resident dendritic cells.

The plasma specimens from ARV-experienced individuals listed in Table 1 who were not virologically suppressed (IDs 6, 7 and 9) further revealed that HIV drug resistance mutations were not uniform across source cell types. Here again, some cellular sources within individuals were associated with wildtype virus while other cellular markers were associated with variants containing one or more drug resistance mutations. The wide diversity of cell types contributing to captured virion in these chronic infections were broadly represented by both lymphoid and myeloid lineage markers. Alternatively, the captured HIV sequences from individual SH874, from whom the specimen was collected soon after seroconversion, presented a rather homogeneous plasma virus population among the different captures with only nominal single nucleotide polymorphisms identified.

## Phylogenetic analysis of captured virion sequences

Phylogenetic analysis of HIV reverse transcriptase (RT) sequences from captured virion represented in Table 1 was performed using the Maximum Likelihood method based on the Tamura 3-parameter model [34, 35] in MEGA v6. Example phylogenetic trees of intra-participant virus sequences among the different captures are shown in Fig 7a–7d. In this analysis, the sequences were stripped of their drug resistance mutations to assess virion backbone sequence variability. We observed that sequences from virions that were CD3$^-$ but CD21$^+$, CD45RA$^+$ or CD45RO$^+$, which could include sources such as immune-resident precursor monocytes (*e.g.*, intra-splenic), myeloid DCs and plasmacytoid DC, respectively, had the greatest sequence distance from CD3$^+$ lymphocyte-derived and high-abundance HLA-DR$^+$ virion sequences from the same sample. The high density of HLA-DR on multiple activated cell types is reflected in the high ambiguity of blood-derived virions carrying this marker. This ambiguity, however, is modulated by the position of the HLA-DR capture in the algorithm series. In the aforementioned early-acute viruses of ID 9012 the pairwise distances were as much as 5.2% between the RT segments. When drug resistance mutations were added back, the RT sequences of captured

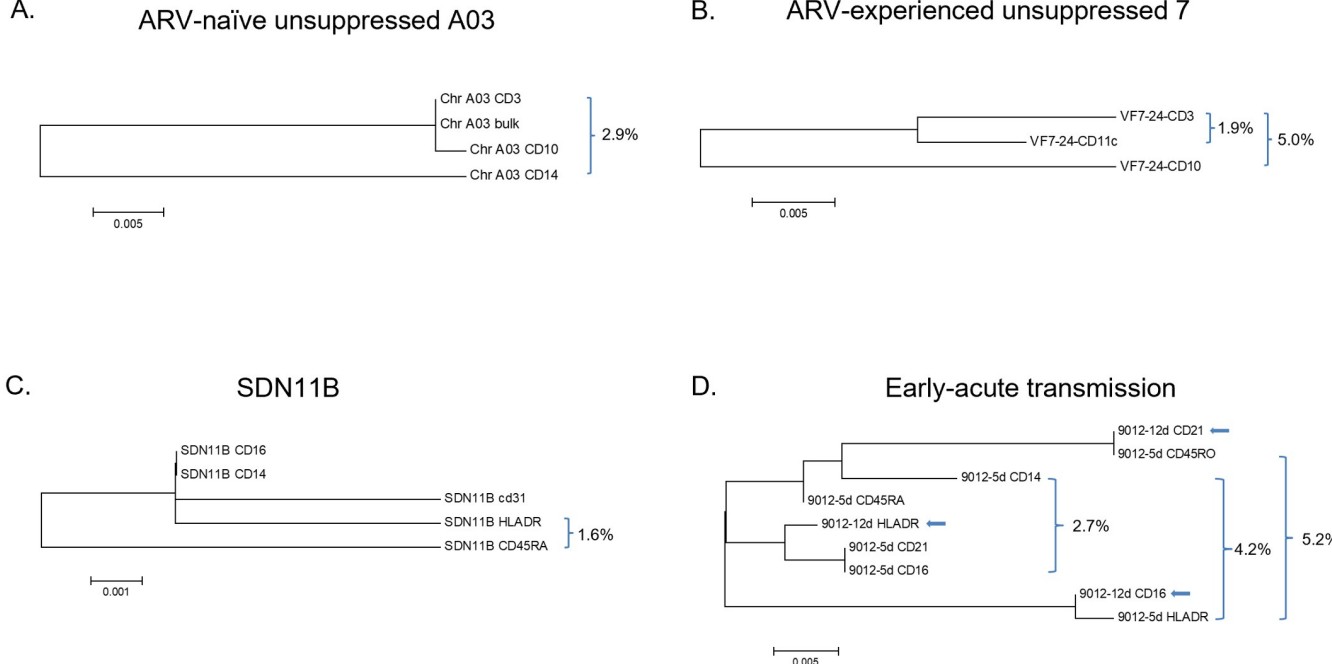

**Fig 7. Maximum likelihood trees illustrating HIV RT sequence differences among captured virions from a representation of individuals shown in Table 1.** Analyses was based on the Tamura 3-parameter model with branch lengths measured in the number of substitutions per site after stripping of drug resistance mutations. Evolutionary analyses were conducted in MEGA6. The clinical circumstance of each individual was: A. treatment-naive with chronic infection; B. ARV-treated but experiencing lack of virologic suppression; C. maternal sequences six weeks post administration of intrapartum single-dose nevirapine (SD-NVP); D. early acute transmission timepoints at 12 days (arrows) and 5 days prior to seroconversion. Percent values are the pairwise genetic distances between the bracketed virions captured by the indicated source markers.

virion had distances as high as 6.4%. At this very early stage of virus expression, myeloid-derived CD16$^+$ virion sequence differed from CD16$^-$/CD14$^+$ virions by >2%, demonstrating multiple myeloid-derived viruses during the earliest days of infection. Virion sequences from CD16$^+$ particles, consistent with expression from homing monocytes and/or tissue-resident macrophages, and HLA-DR$^+$ virion that were suggestive of activated T cells and dendritic cells had the greatest nucleotide ambiguity. For all examples shown, the RT region genetic distance between the non-lymphocyte and the majority lymphocyte-derived virions within individuals was greater than 1.5%.

## Discussion

Through an empirical immunocapture process using mAbs against cell-source markers embedded in virion envelopes we separated HIV variants expressed in blood and evaluated the genomic RNA from each virion capture step. By both interrogating the choice of mAb targets and immunocapture order the iterative algorithms allowed us to improve segregation of HIV into virion pools of reduced genome ambiguity to better explore the genetic variants associated with different cell source markers. While the primary aim at each step in the algorithm design was to narrow the possible cellular origins of HIV particle subpopulations, we were additionally interested in whether we would be able to uncover occult variant expression undetected by typical laboratory methods. The ultimate goal was to gain confidence in the specificity of captures to allow assessments of unique virus sub-populations. The capture sequences presented herein largely explored virion of CD3+ T cell origin together as a group then enriched for

sources of myeloid-derived subsets, followed by virion from other differentiation/maturation stages or cell types that were not captured by the upstream markers.

The diversity of variants that were uncovered in blood not only revealed differences in background polymorphisms between virions recovered from the capture steps but also notable differences in drug resistance mutations. Since drug resistance mutations in the absence of antiretroviral drug pressure are disadvantageous to replicative fitness, their presence above natural quasispecies frequencies provided further evidence that these virions differentially arose from distinct cellular reservoirs. Observing different DRMs when sampling a very brief infection timeframe provides additional credence to rapid compartmental diversification. In all, the findings revealed that virion expression varies with the course of each individual's infection and antiretroviral drug experience. Other than observing a coalescence of variant expression soon after seroconversion, there was no predictable pattern of cell source expression, which speaks to the variability of virus adaption in each host.

The ability of captures to extricate variants that were not identified by sensitive mutation-specific PCR or NGS suggests an inherent hurdle in the ability of conventional methods to successfully identify variants at stoichiometrically low copies within the greater viral quasispecies. The minority level drug-resistant variants not identified by prior bulk sequencing often carried cell markers primarily consistent with homing monocytes and with classical monocytes and macrophages (CD3$^-$ CD16$^+$ and/or CD14$^+$). The clinical relevance of drug-resistant viruses in myeloid cell lineages is not clear but may help to explain why low-level detection of drug resistance might not always result in poor virologic suppression. The role of the complex spectrum of myeloid cells in viral persistence requires substantially more research.

We saw in longitudinal specimens from an early-acute infection a very rapid transfer and turnover of variant genotypes between cell lineages prior to a dominant genotype becoming established. We found that the earliest expressed variants in blood were predominantly of lymphocyte origin and had quickly made their way into myeloid cell populations prior to the lymphocyte-derived genotype transitioning to wildtype. The very high genome diversity in captured virion from the pre-seroconversion timepoints recapitulated our earlier report on the presence of disparate quasispecies observed during the earliest days of infection [33]. Furthermore, the rapid loss of drug resistance mutations after transmission reflects the often-deleterious nature of these mutations when no longer under antiretroviral drug pressure after entry into the new host. This genomic condensation to what appears as a mono-phyletic population by the time of seroconversion matches the common observation of a single variant at diagnosis, as people are typically diagnosed after seroconversion. Additionally, the high genetic distance observed with some captures may reflect evidence of superinfections which we have previously shown in longitudinal assessments may not be an infrequent occurrence [36].

Among clinical specimens from established HIV infections, we found that CD45RO$^+$ and CD21$^+$ virions, here suggestive of DC lineages, had relatively higher viral genome divergence from plasma bulk sequences than did other capture fractions. These may reflect a differentially-evolving reservoir or possibly early myeloid seeding by variants that have largely disappeared from blood, as was described above for the early-acute infection. Of note, CD1c + myeloid DCs (mDC), expressing myeloid antigens CD11c and CD45RO, are the major population of human mDCs in blood, tissues, and lymphoid organs, comprising approximately 1% of mononuclear cells [37]. The high nucleotide ambiguity observed in the lymphoid HLA-DR$^+$ and myeloid CD16$^+$ virions would suggest that cell types presenting these markers contribute to multiple variants which are expressing, intermixing, and possibly evolving at higher rates.

The use of the immunocapture strategies presented here may only be scratching the surface of even more complex HIV expression that lies deeper still. Marker combinations for additional deep tissue sources not represented in peripheral blood samples would require intensive investigation. Since the antibody selection to create iterative algorithms is dynamic the use of mAbs to reach the cell source populations of interest may require considerable design to obtain optimized capture schemes. For example, anti-CD123, instead of anti-CD45RA, could be used in the series illustrated in Fig 2 to capture plasmacytoid DC variants, but it would depend on the granularity desired for those populations and what additional markers to consider.

A few limitations are evident as the capture assay relies on the ability to bind targets embedded within intact virus particle envelopes. Length of storage and multiple freeze-thaws of remnant plasma samples were factors that affected the ability to capture virions. Evidence of plasma sample integrity could be gleaned from the amplification success of captured HLA-DR$^+$ virions, since HLA-DR is expressed on activated cell types which are typically the source of the vast majority of expressed virus in blood. Another limitation is that particle subpopulations that have a particularly low host-protein surface density or are present at very low copies may not be adequately represented at the time of sampling to allow for genetic analysis. It is also important to note that *in vitro* virus culture can alter cell surface marker presentation and may not necessarily represent the breadth of targets present in clinical specimens. We also acknowledge that the broad diversity of cell activation and maturation states at the time the blood is sampled affects the ability to determine relative capture efficiency for a particular marker.

A potential confounder in targeting markers indicative of antigen presenting cell sources, for example anti-CD21, is that the captures might immunoprecipitate cellular debris of CD protein bound to virus, thus those genomes may not necessarily reflect virus particles *originating* from presenting cells. However, evidence in support of presenting cells being able to serve as a competent source was the finding that CD21$^+$ variants had genetic sequences that were often distinct from CD3$^+$ particles, suggesting they were not simply lymphocyte-derived particles sequestered in presenting cell-virus complexes. Additionally, if CD3+CD14+ T cell-monocyte complexes ([38] see S2 Fig) are expressing virus, those virions may contain both lymphoid and myeloid markers and could be interpreted as the same variant originating from two different sources.

In summary, virion particle immunocapture algorithms applied to blood plasma specimens uncovered HIV subpopulations associated with different cell types that were more complex than what could be discerned by sensitive genotyping and mutation-specific (low-frequency) PCR. Distinct wildtype and drug-resistant variants associated with different cell lineage markers revealed that HIV pools can adapt their genomes, sometimes remarkably quickly, in response to environmental influences such as antiretroviral drug pressures. Because virion immunocapture can draw out low-abundance subpopulations that might not be successfully identified by conventional detection methods, it is a powerful tool to help identify concealed virus sources, including sources resistant to therapy, that serve as persistent reservoirs of HIV. Further examination of these viral subpopulations may aid in a better understanding of HIV persistence under highly active ART and inform strategies for durable virus suppression. Moreover, the ability to expand identification of cellular sources of re-emergent viruses stimulated out of latency could provide valuable information on targets to eradicate HIV. While the findings described here are from blood plasma, other body fluids are also being explored using this method in order to elucidate the relationships between viruses in peripheral blood and in other fluid compartments.

## Supporting information

**S1 Fig. Test of virus particle recovery after column washes with and without Tween 20 on anti-HLA-DR and anti-CD16 captures of DNase-treated cultured HIV$_{ADA-M}$ virus.** RNA amplifications shown for anti-HLA-DR columns (green lines) and anti-CD16 columns (blue lines), and column with no antibody present (red line). Curves shown for columns either washed three times without Tween 20 or washed with buffer containing 1% Tween 20 as per protocol. Amplification from the no-Ab column (red line) demonstrates particles are non-specifically retained on columns in the absence of Tween 20.
(TIF)

**S2 Fig. An example sequence alignment of HIV captures within a plasma sample when altering the position of the CD3$^+$ capture.** The upper and lower alignments represent two regions in HIV-1 reverse transcriptase (RT) from an individual who was administered intrapartum SD-NVP. Two series of captures are demonstrated in the alignments. In the initial capture order the anti-CD3 capture was placed after anti-HLA-DR (CD3 post-DR), then on reanalysis the CD3 capture was placed at the beginning of the algorithm (CD3 start). The "CD3 start" would have comprised a broad representation of T cell-derived virion, whereas placed after HLA-DR the CD3 capture would have represented non-activated, mature T cell sources. RT amino acid codon positions are indicated above. Codon 101 (A→G) and 103 (A→C) mutations confer NVP resistance and were differentially selected among virions associated with different CD markers as were background polymorphisms (*e.g.*, codon 162).
(TIF)

**S3 Fig.** Comparison of HIV mutations at codons associated with drug resistance identified by next-generation sequencing (NGS) (A.) and those obtained from virion particle captures (B.) from the early-acute sample ID 9012 collected at 12 days pre-seroconversion. A. Trimmed NGS reads were mapped to HXB2 reverse transcriptase (RT) reference sequence followed by the removal of any duplicate mapped reads. Variants above a 2% frequency of total read coverage are included. B. Captured drug resistance mutations at this timepoint that were not identified by NGS (in red) were all later identified by captures at 5 days pre-seroconversion associated with myeloid populations.
(TIF)

**S1 Table. Test of percent Tween 20 on non-specific retention of virus on columns without antibody.**
(DOCX)

**S2 Table. Monoclonal antibodies used in capture evaluations.**
(DOCX)

**S1 File. Immunocapture of virion from body fluids.**
(PDF)

## Acknowledgments

We thank Dr. Alison Swaims-Kohlmeier for training on cell sorting and Dawn Little for additional assistance with flow cytometry. HIV-1$_{IIIB}$ was obtained through the NIH AIDS Reagent Program, Division of AIDS, NIAID, NIH: HIV-1 IIIB Virus from Dr. Robert Gallo (cat# 398). HIV-1 ADA-M Virus from Dr. Howard Gendelman was obtained through the NIH AIDS Reagent Program, Division of AIDS, NIAID, NIH.

**Disclaimer**: The findings and conclusions of this manuscript are those of the authors and do not necessarily represent the official views of the Centers for Disease Control and Prevention.

## Author Contributions

**Conceptualization:** Jeffrey A. Johnson.

**Data curation:** Sarah Sabour, Jonathan T. Lipscomb, Ariana P. Santos Tino, Jeffrey A. Johnson.

**Formal analysis:** Sarah Sabour, Jin-fen Li, Jeffrey A. Johnson.

**Investigation:** Sarah Sabour, Jin-fen Li, Jonathan T. Lipscomb, Ariana P. Santos Tino.

**Methodology:** Jeffrey A. Johnson.

**Project administration:** Jeffrey A. Johnson.

**Resources:** Jeffrey A. Johnson.

**Supervision:** Jeffrey A. Johnson.

**Validation:** Sarah Sabour, Jin-fen Li, Jeffrey A. Johnson.

**Writing – original draft:** Sarah Sabour, Jeffrey A. Johnson.

**Writing – review & editing:** Jeffrey A. Johnson.

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
