## [Decision Letter · Decision Letter 0]

8 Oct 2023

PONE-D-23-23838Immunocapture of Cell Surface Proteins Embedded in HIV Envelopes Uncovers Considerable Virion Genetic Diversity Associated with Different Source Cell TypesPLOS ONE

Dear Dr. Johnson,

Thank you for submitting your manuscript to PLOS ONE. After reviewing the article, it was deemed well-written and presented. However, a few technical issues require attention and revision. After considering the reviewer's suggestions, it was determined that improvements are necessary due to the precise protocol-based nature of this articles.Therefore, we invite you to submit a revised version of the manuscript that addresses the points raised during the review process.

ACADEMIC EDITOR:This is an excellent article. A study on HIV particles in the bloodstream identified distinct variants linked to different cell types. A new method for capturing the virus was developed, allowing the detection of previously undetected drug-resistance mutations. This has implications for HIV cure strategies and understanding genetic diversity in other viral infections. However, after reviewing the article, it was deemed well-written and presented. However, a few technical issues require attention and revision. After considering the reviewer's suggestions, it was determined that improvements are necessary due to the precise protocol-based nature of this article.

We look forward to receiving your revised manuscript.

Kind regards,

Satish Rojekar, Ph.D.

Academic Editor

PLOS ONE

5. We note you have not yet provided a protocols.io PDF version of your protocol and/or a protocols.io DOI. When you submit your revision, please provide a PDF version of your protocol as generated by protocols.io (the file will have the protocols.io logo in the upper right corner of the first page) as a Supporting Information file. The filename should be S1_file.pdf, and you should enter “S1 File” into the Description field. Any additional protocols should be numbered S2, S3, and so on. Please also follow the instructions for Supporting Information captions [https://journals.plos.org/plosone/s/supporting-information#loc-captions]. The title in the caption should read: “Step-by-step protocol, also available on protocols.io.”

Please assign your protocol a protocols.io DOI, if you have not already done so, and include the following line in the Materials and Methods section of your manuscript: “The protocol described in this peer-reviewed article is published on protocols.io (https://dx.doi.org/10.17504/protocols.io.[...]) and is included for printing purposes as S1 File.” You should also supply the DOI in the Protocols.io DOI field of the submission form when you submit your revision.

If you have not yet uploaded your protocol to protocols.io, you are invited to use the platform’s protocol entry service [https://www.protocols.io/we-enter-protocols] for doing so, at no charge. Through this service, the team at protocols.io will enter your protocol for you and format it in a way that takes advantage of the platform’s features. When submitting your protocol to the protocol entry service please include the customer code PLOS2022 in the Note field and indicate that your protocol is associated with a PLOS ONE Lab Protocol Submission. You should also include the title and manuscript number of your PLOS ONE submission.

Additional Editor Comments:

After reviewing the article, it was deemed well-written and presented. However, a few technical issues require attention and revision. After considering the reviewer's suggestions, it was determined that improvements are necessary due to the precise protocol-based nature of this articles.

I recommend that an author revise the manuscript and submit an improved version.

Reviewer#1

In the article the authors have used different target specific mAb to segregate the virus particles of the different origins. They have used the serial method where samples are allowed to be attached to the selected mAb one after another in sequence. The authors have done an excellent job of designing the study and explaining the results and conclusions.

As the reviewer of the paper, I have the following questions for the authors -

1. In the line 89, how did the authors determine the amount of antibody-bead material in the column was sufficient to support binding of up to 500,000 virions.

2. From the line 100, can the authors give more details about the Different wash buffer formulations evaluated to increase capture specificity?

3. Can the authors please provide the information about the number of samples of human origin and number of people the samples originated from.

4. Can the authors expand on the results of the study in regard to the following points

a. How similar is the distribution of the virus particles from different subjects?

b. what type of cells harbored the highest variety of virus particles?

5. I would request the authors to make the format of the references uniform.

Reviewer#2

However, the author may take note of the major and minor remarks listed below to improve the manuscript:

I have major concerns about discussion and methodology. I would encourage authors to rewrite the discussion with more references, as in the current state it seems to be just observations. Also, though the article type is laboratory protocol, I recommend adding more references in the methodology section as these protocols are complex.

Abstract:

The abstract is written very well, and it provides a concise overview of the study's objectives and findings.

Introduction:

The introduction needs to be refined, as it is very short on study background. It should effectively introduce the key concepts, research gaps, and objectives of the study.

Here are a few suggestions for improvement:

• I kindly request that authors emphasize how HIV persistence and drug resistance arise based on drug permeability and cell lineages.

• What are the major ‘research gaps’ that lead to the study hypothesis?

Material and methods:

This section provides a solid foundation for conducting the study. However, in this article, some areas could be further improved to enhance clarity and reproducibility.

• As mentioned earlier, at certain points, there is a need to add more references. For example, Line No. 82,

• I am not able to find the number of samples used in this study. Please mention them in this section.

• Is there any way to include more information about wash buffer formulation as the whole protocol and study are based on this phenomenon?

• Please remove the section that starts from line 153 and keep it at the start of the methodology section.

• The part written from lines 161 to 171 can be shortened as it seems redundant to some parts of the introduction.

• Does the use of lysis buffer on the column to recover RNA hamper the column's capacity to reuse it again for the next elution?

Results:

The overall result section is written very well.

• Please change figure 1.

• Also, the authors have given the cell sorting data as supplementary. However, I personally feel putting these results in a manuscript will strengthen the study hypothesis and the value of the manuscript.

Discussion:

• As mentioned earlier, I recommend adding more references to strengthen the observed results. The conclusion needs to be rewritten as it lacks conciseness about the observed parameters and the importance of the study.

• There is scope for shortening the summary.

Reviewers' comments:

Reviewer's Responses to Questions

**Comments to the Author**

1. Does the manuscript report a protocol which is of utility to the research community and adds value to the published literature?

Reviewer #1: Yes

Reviewer #2: Yes

2. Has the protocol been described in sufficient detail?

To answer this question, please click the link to protocols.io in the Materials and Methods section of the manuscript (if a link has been provided) or consult the step-by-step protocol in the Supporting Information files.

The step-by-step protocol should contain sufficient detail for another researcher to be able to reproduce all experiments and analyses.

Reviewer #1: Yes

Reviewer #2: Yes

3. Does the protocol describe a validated method?

Reviewer #1: Yes

Reviewer #2: Yes

4. If the manuscript contains new data, have the authors made this data fully available?

Reviewer #1: Yes

Reviewer #2: Yes

**5. Is the article presented in an intelligible fashion and written in standard English?**

Reviewer #1: Yes

Reviewer #2: Yes

6. Review Comments to the Author

Reviewer #1: In the article the authors have used different target specific mAb to segregate the virus particles of the different origins. They have used the serial method where samples are allowed to be attached to the selected mAb one after another in sequence. The authors have done an excellent job of designing the study and explaining the results and conclusions.

As the reviewer of the paper, I have the following questions for the authors -

1. In the line 89, how did the authors determine the amount of antibody-bead material in the column was sufficient to support binding of up to 500,000 virions.

2. From the line 100, can the authors give more details about the Different wash buffer formulations evaluated to increase capture specificity?

3. Can the authors please provide the information about the number of samples of human origin and number of people the samples originated from.

4. Can the authors expand on the results of the study in regard to the following points

a. How similar is the distribution of the virus particles from different subjects?

b. what type of cells harbored the highest variety of virus particles?

5. I would request the authors to make the format of the references uniform.

Reviewer #2: However, the author may take note of the major and minor remarks listed below to improve the manuscript:

I have major concerns about discussion and methodology. I would encourage authors to rewrite the discussion with more references, as in the current state it seems to be just observations. Also, though the article type is laboratory protocol, I recommend adding more references in the methodology section as these protocols are complex.

Abstract:

The abstract is written very well, and it provides a concise overview of the study's objectives and findings.

Introduction:

The introduction needs to be refined, as it is very short on study background. It should effectively introduce the key concepts, research gaps, and objectives of the study.

Here are a few suggestions for improvement:

• I kindly request that authors emphasize how HIV persistence and drug resistance arise based on drug permeability and cell lineages.

• What are the major ‘research gaps’ that lead to the study hypothesis?

Material and methods:

This section provides a solid foundation for conducting the study. However, in this article, some areas could be further improved to enhance clarity and reproducibility.

• As mentioned earlier, at certain points, there is a need to add more references. For example, Line No. 82,

• I am not able to find the number of samples used in this study. Please mention them in this section.

• Is there any way to include more information about wash buffer formulation as the whole protocol and study are based on this phenomenon?

• Please remove the section that starts from line 153 and keep it at the start of the methodology section.

• The part written from lines 161 to 171 can be shortened as it seems redundant to some parts of the introduction.

• Does the use of lysis buffer on the column to recover RNA hamper the column's capacity to reuse it again for the next elution?

Results:

The overall result section is written very well.

• Please change figure 1.

• Also, the authors have given the cell sorting data as supplementary. However, I personally feel putting these results in a manuscript will strengthen the study hypothesis and the value of the manuscript.

Discussion:

• As mentioned earlier, I recommend adding more references to strengthen the observed results. The conclusion needs to be rewritten as it lacks conciseness about the observed parameters and the importance of the study.

• There is scope for shortening the summary.

7. PLOS authors have the option to publish the peer review history of their article (what does this mean?). If published, this will include your full peer review and any attached files.

Reviewer #1: No

Reviewer #2: **Yes: **Namdev S Togre

---

## [Author Response · Author response to Decision Letter 0]

31 Oct 2023

Reviewer #1: In the article the authors have used different target specific mAb to segregate the virus particles of the different origins. They have used the serial method where samples are allowed to be attached to the selected mAb one after another in sequence. The authors have done an excellent job of designing the study and explaining the results and conclusions.

As the reviewer of the paper, I have the following questions for the authors -

1. In the line 89, how did the authors determine the amount of antibody-bead material in the column was sufficient to support binding of up to 500,000 virions.

Response: Additional detail has been added to this section to explain the amount of Ab-bead complex can easily accommodate 500,000 virion copies; but we found that using higher copies occludes the column to cause spurious amplification. Hence, we limit the input to no more than 500,000 copies.

2. From the line 100, can the authors give more details about the Different wash buffer formulations evaluated to increase capture specificity?

Response: We have included a prepared supplemental table to illustrate the evaluation if it is helpful. We assessed both fetal bovine serum and bovine serum albumin as the block protein and determined that BSA was better for alleviating non-specific column retention. The additional assessment was the percent Tween 20 to use as we were concerned with the balance between clean columns and potential virus disruption. 

3. Can the authors please provide the information about the number of samples of human origin and number of people the samples originated from.

Response: This information was added to the paragraph on human subjects. A definitive number cannot be recalled as hundreds of clinical samples were used in the R&D over the decade.

4. Can the authors expand on the results of the study in regard to the following points

a. How similar is the distribution of the virus particles from different subjects?

Response: The general observation is that there are no predictable patterns between individuals and have now discussed this in line 419.

b. what type of cells harbored the highest variety of virus particles?

Response: While we mentioned this information in the Discussion on line 443, we were remiss in including this in the Results. Often, for blood, HLA-DR+ particles had higher ambiguity due to its associated with multiple activated cell types. We have added our observation on line 378. 

5. I would request the authors to make the format of the references uniform.

Response: Thank you for your observation. The formatting has been made uniform.

Reviewer #2: However, the author may take note of the major and minor remarks listed below to improve the manuscript:

I have major concerns about discussion and methodology. I would encourage authors to rewrite the discussion with more references, as in the current state it seems to be just observations. Also, though the article type is laboratory protocol, I recommend adding more references in the methodology section as these protocols are complex.

Abstract:

The abstract is written very well, and it provides a concise overview of the study's objectives and findings.

Introduction:

The introduction needs to be refined, as it is very short on study background. It should effectively introduce the key concepts, research gaps, and objectives of the study.

Here are a few suggestions for improvement:

• I kindly request that authors emphasize how HIV persistence and drug resistance arise based on drug permeability and cell lineages.

• What are the major ‘research gaps’ that lead to the study hypothesis?

Response: Thank you for the suggestion. Reframing of the aims and issues as well as additional references have been provided in the Introduction (please see highlighted text)

Material and methods:

This section provides a solid foundation for conducting the study. However, in this article, some areas could be further improved to enhance clarity and reproducibility.

• As mentioned earlier, at certain points, there is a need to add more references. For example, Line No. 82,

Response: Additional reference added for this line and in the Intro.

• I am not able to find the number of samples used in this study. Please mention them in this section. 

Response: (also provided above) This information was added to the paragraph on human subjects. A definitive number cannot be recalled as hundreds of clinical samples were used in the R&D over the decade.

• Is there any way to include more information about wash buffer formulation as the whole protocol and study are based on this phenomenon?

Response: (also provided above) We have included a prepared supplemental table to illustrate the evaluation if it is helpful. We assessed both fetal bovine serum and bovine serum albumin as the block protein and determined that BSA was better for alleviating non-specific column retention. The additional assessment was the percent Tween 20 to use as we were concerned with the balance between clean columns and potential virus disruption. 

• Please remove the section that starts from line 153 and keep it at the start of the methodology section.

Response: Done

• The part written from lines 161 to 171 can be shortened as it seems redundant to some parts of the introduction.

Response: Thank you for the suggestion. The text has been reduced.

• Does the use of lysis buffer on the column to recover RNA hamper the column's capacity to reuse it again for the next elution?

Response: Each Ab-bead complex is assigned its own fresh column. Apologies that this detail was not clear, and we have added language on line 122 that we hope clarifies.

Results:

The overall result section is written very well.

• Please change figure 1.

Response: We are not clear as to what this reviewer recommends as far as changing the figure. However, the figure is now condensed to hopefully better illustrate the representative steps.

• Also, the authors have given the cell sorting data as supplementary. However, I personally feel putting these results in a manuscript will strengthen the study hypothesis and the value of the manuscript.

Response: Thank you for this suggestion. We were concerned with the number of figures but are happy to move this figure to be more prominently represented (new Fig 4).

Discussion:

• As mentioned earlier, I recommend adding more references to strengthen the observed results. The conclusion needs to be rewritten as it lacks conciseness about the observed parameters and the importance of the study.

Response: Additional references have been added.

• There is scope for shortening the summary.

Response: The introductory paragraph of the Discussion has been shortened. However, additional observations have been added in the Discussion to address reviewer suggestions.

---

## [Decision Letter · Decision Letter 1]

20 Dec 2023

Immunocapture of Cell Surface Proteins Embedded in HIV Envelopes Uncovers Considerable Virion Genetic Diversity Associated with Different Source Cell Types

PONE-D-23-23838R1

Dear Dr. Jeffrey Johnson, 

We’re pleased to inform you that your manuscript has been judged scientifically suitable for publication and will be formally accepted for publication once it meets all outstanding technical requirements.

Kind regards,

Satish Rojekar, Ph.D.

Academic Editor

PLOS ONE

Additional Editor Comments (optional):

Thank you for submitting your manuscript to PLOS ONE. The author has addressed most of the reviewer's comments. Based on my assessment, I am pleased to inform you that your manuscript has been accepted for publication in its present form.

Reviewers' comments:

Reviewer's Responses to Questions

**Comments to the Author**

1. Does the manuscript report a protocol which is of utility to the research community and adds value to the published literature?

Reviewer #2: Yes

2. Has the protocol been described in sufficient detail?

To answer this question, please click the link to protocols.io in the Materials and Methods section of the manuscript (if a link has been provided) or consult the step-by-step protocol in the Supporting Information files.

The step-by-step protocol should contain sufficient detail for another researcher to be able to reproduce all experiments and analyses.

Reviewer #2: Yes

3. Does the protocol describe a validated method?

Reviewer #2: Yes

4. If the manuscript contains new data, have the authors made this data fully available?

Reviewer #2: N/A

**5. Is the article presented in an intelligible fashion and written in standard English?**

Reviewer #2: Yes

6. Review Comments to the Author

Reviewer #2: Thank you, author for adding the necessary changes. This manuscript can be accepted for publication in present form.

7. PLOS authors have the option to publish the peer review history of their article (what does this mean?). If published, this will include your full peer review and any attached files.

Reviewer #2: **Yes: **Namdev S Togre

---

## [Editor Report · Acceptance letter]

17 Feb 2024

PONE-D-23-23838R1 

PLOS ONE

Dear Dr. Johnson, 

I'm pleased to inform you that your manuscript has been deemed suitable for publication in PLOS ONE. Congratulations! Your manuscript is now being handed over to our production team.

Kind regards, 

on behalf of

Dr. Satish Rojekar 

Academic Editor

PLOS ONE